# Effects of Unloaded vs. Ankle-Loaded Plyometric Training on the Physical Fitness of U-17 Male Soccer Players

**DOI:** 10.3390/ijerph17217877

**Published:** 2020-10-27

**Authors:** Mehrez Hammami, Nawel Gaamouri, Katsuhiko Suzuki, Ridha Aouadi, Roy J. Shephard, Mohamed Souhaiel Chelly

**Affiliations:** 1Research Unit (UR17JS01) «Sport Performance, Health & Society», Higher Institute of Sport and Physical Education of Ksar Saîd, University of “La Manouba”, 2010 Tunis, Tunisia; nawel.gaamouri@yahoo.fr (N.G.); ridha_aouadi@yahoo.fr (R.A.); csouhaiel@yahoo.fr (M.S.C.); 2Department of Biological Sciences Applied for Physical Activities and Sport, Higher Institute of Sport and Physical Education of Ksar Said, University of “La Manouba”, 2010 Tunis, Tunisia; 3Faculty of Sport Sciences, Waseda University, Tokorozawa 359-1192, Japan; katsu.suzu@waseda.jp; 4Faculty of Kinesiology and Physical Education, University of Toronto, Toronto, ON M5S, Canada; royjshep@shaw.ca

**Keywords:** stretch-shortening cycle, additional weight, ability-to-change-direction, speed, balance, repeated change of direction

## Abstract

The aim of this study was to compare the impact of two differing plyometric training programs (loaded plyometrics (with 2.5% of body mass placed above the ankle joint) vs. unloaded plyometrics), performed biweekly for 10 weeks, on the physical fitness of elite junior male soccer players. Participants aged 16.0 ± 0.5 years were randomly assigned between unloaded plyometrics (UP; *n* = 12), loaded plyometrics (LP; *n* = 14) and control (C; *n* = 12) groups. Two-way analyses of performance (group x time) were assessed by 40-m sprint times; 9–3–6–3–9 m sprints with 180° turns (S180°); 9–3–6–3–9 m sprints with backward and forward running (SBF); and 4 × 5 m sprints (S4 × 5 m); four jump tests; measures of static and dynamic balance; repeated change of direction tests and the Yo-Yo intermittent recovery test. Both LP and UP enhanced sprinting performance relative to C (*p* < 0.05) but performance increased more in LP relative to UP (*p* < 0.05) in all sprints except 40 m. Change of direction times were also significantly shortened by LP relative to UP (*p* < 0.05) and C (*p* < 0.01) in all tests, with no significant differences between UP and C. Jumps heights increased similarly in both LP and UP relative to C (*p* < 0.05), with no significance between LP and UP. LP and UP also enhanced repeated change of direction scores relative to C (*p* < 0.01) with greater changes in LP than in UP (*p* < 0.01). Finally, LP enhanced some balance scores relative to UP (*p* < 0.05) and C (*p* < 0.05). We conclude that the introduction of 10 weeks of in-season loaded plyometrics into the regimen of U17 male soccer players yields gains in several physical performance scores relative to either unloaded plyometrics or the control training regimen.

## 1. Introduction

Soccer players need a combination of strength and power to win ball possession, score, and assist or prevent goals [1,2]. Among tactics adopted to improve their jumping and sprinting abilities [3,4], plyometric training has proven one of the more effective ways of improving the rate of force development, sprinting and jumping [5,6,7,8,9]. Plyometric training is an exercise in which an eccentric muscle contraction is quickly followed by a concentric muscle contraction [10,11,12]. This is a type of strength training, and it is currently very popular in training for many sports such as soccer, handball, and basketball [5,13,14]. It involves performing bodyweight jumping-type exercises and throwing medicine balls, exploiting the so-called stretch–shortening cycle (SSC) of muscle action [5]. The SSC enhances the ability of the neural and musculotendinous systems to produce maximal force in the shortest amount of time, prompting the use of plyometric exercise as a bridge between strength and speed [15]. In this regard, plyometric exercises has been used extensively as a method of augmenting dynamic athletic actions such as sprinting, change of direction and jumping performance [5,13,14]. Several articles have discussed the use of plyometric training with additional external loading. Thus, Cronin et al. [16] demonstrated larger improvements in jumping performance when using this approach, attributing this to greater ground reaction forces. Rosas et al. [17] also compared the effects on jumping with or without haltere-type hand-held loading; after 6 weeks of training, gains in both vertical and horizontal jumping ability were greater for the loaded group. However, Kobal et al. [18] found that neither type of plyometric training yielded worthwhile improvements in maximal speed and power, blaming this poor response on interference from concurrent activities. All of these studies loaded the upper limbs, and none loaded the ankles. Nevertheless, this practice has been adopted in rehabilitation, and by those seeking to lose weight or increase other aspects of sports performance. Today, many people do not hesitate to use weights from 500 g to several kilos when skipping, jogging, and even playing basketball or tennis.

Few previous studies have examined the impact of combined forms of plyometrics on soccer players [17,18], and this is the first report to have examined the effect of ankle-loaded plyometric training (2.5% of body mass) upon repeated change of direction ability. Our study compared loaded versus unloaded plyometric training in terms of gains in sprinting performance, ability to change direction repeatedly, static and dynamic balance, vertical jumping, and maximal aerobic power. It was hypothesized that there would be a superior effect of loaded as compared to unloaded plyometric training.

## 2. Materials and Methods

### 2.1. Ethical Approval

All procedures were approved by the Institute’s Committee on Research for the Medical Sciences (Manouba University Ethics Committee: UR17JS01) and performed in accordance with the current national laws and regulations and the Declaration of Helsinki. Informed consent was gained from all participants and their parents or guardians after a verbal and a written explanation of the experimental protocol and its potential risks and benefits. Participants were assured that they could withdraw from the trial without penalty at any time.

### 2.2. Participants

The study aimed to compare the effects of 10 weeks of biweekly unloaded versus loaded (2.5% of body mass, placed above the ankle joint) plyometric training on sprinting, jumping, ability to change direction, balance, repeated changes of direction, and aerobic power in elite junior male soccer players. Using the PEDRO scale [19], a group of experienced elite male soccer players (38 players; 3 goalkeepers and 35 field players) was assigned randomly (Table 1) between unloaded plyometric training (UP; n = 12; age = 16.2 ± 0.2 years; body mass = 59.8 ± 2.8 kg; height = 1.78 ± 0.21 m; body fat = 10.7 ± 3.0%), loaded plyometric training (LP; n = 14; age = 16.3 ± 0.4 years; body mass = 60.9 ± 3.4 kg; height = 1.77 ± 0.31 m; body fat = 11.1 ± 2.1%), and controls (who continued with the standard in-season regimen) (C; n = 12; age = 16.4 ± 0.2 years; body mass = 58.9 ± 3.7 kg; height = 1.78 ± 0.32 m; body fat = 10.4 ± 2.6%). Each group contained one goalkeeper and the remainder were field players. The maturity status of each participant was calculated as a maturity offset [20]:Maturity Offset = −9.236 + 0.000278 leg length × sitting height −0.001663 age × leg length + 0.007216 age × sitting height + 0.02292 weight × height (years).(1)

The age at peak height velocity was 14.1 ± 0.3, 14.1 ± 0.4 and 14.2 ± 0.4 years for LP, UP and C respectively. All three groups (LP, UP and CG) belonged to the same football team. All participants were engaged in soccer training 6–7 times per week, played one official game per week from the beginning of the competitive season (September) until the end of the trial period (March), and also undertook a weekly two-hour physical education course. Standard training sessions lasted 90–100 min, emphasizing skill activities at various intensities, offensive and defensive strategies, and 25 to 30 min of soccer play with only brief interruptions by the coach. Each Sunday, the team played either an official match (11 and 16 players) or engaged in a friendly match. Thus, all participants engaged in the same duration of weekly training activity.

### 2.3. Experimental Design

Two weeks before the initial experimental measurements, all participants completed two familiarization trials of all procedures except the repeated change of direction test (which was only practiced once) and the anthropometric assessments (which required no familiarization). Figure 1 presents a revised CONSORT diagram of the levels of reporting and explaining participant flow. The initial definitive measurements were made four months into the playing season, during the first two weeks of January (which was a rest period for all players), and tests were repeated after completing the 10-week intervention, 5–9 days after the last training session. On both occasions, participants undertook 40 m sprints, three change of direction tests (sprinting 9–3–6–3–9 m with 180° turns; sprinting 9–3–6–3–9 m with backward and forward running; sprint 4 × 5 m), a repeated change of direction (RCOD) test, four jump tests (squat jump, countermovement jump, countermovement jump with aimed arms and five jump tests); measures of static and dynamic balance, and an estimate of maximal aerobic power.

Before the start of the study, participants had practiced soccer for 8 years and participated in many international tournaments (From U10 to U17). All had achieved a good level of physical preparation at the beginning of the season (seven training sessions per week for 2 months). When the season began (September) we emphasized resistance strength training with light loads (block working for 2 months), then we completed the first phase (first place) and qualified to the play-off phase. The present investigation was undertaken during the second phase of the national championships (January to April).

Testing sessions were carried out at a consistent time of the day, and under the same experimental conditions (tartan surfaced stadium), at least 3 days after the most recent competition. Players maintained their normal intake of food and fluids during assessments. However, they drank no caffeine-containing beverages during the 4 h preceding testing, and ate no food for 2 h. Verbal encouragement was provided by the experimenters throughout.

The 38 elite male soccer players were examined by the team physician, focusing on conditions that might preclude resistance training and all were in good health. The three assigned groups were well-matched in terms of their initial physical characteristics, with no statistically significant inter-group differences.

All participants were engaged in soccer training 6–7 times per week, played one official game per week from the beginning of the competitive season (September) until the end of the trial (March), and also had a weekly two hours of physical education course. Standard training sessions lasted 90–100 min; emphasizing skill activities at various intensities, offensive and defensive strategies, and 25 to 30 min of soccer play with only brief interruptions by the coach.

### 2.4. Details of Standard and Modified Plyometric Training

Details of the two plyometric regimens are given in Table 2. Each Tuesday and Thursday for 10 weeks, the two experimental groups replaced the initial part of their standard training program (10–15% of regular soccer training times) with plyometric exercises (vertical and horizontal directions) in stable surface, intended to yield an optimal increase in muscle strength, followed by an increase in muscle power. Programs comprising four principal exercises included hopping, hurdles and horizontal jumps, finishing with a 20 m sprint. Each exercise set comprised 6 jumps (6 contacts) and a sprint of 20 m [14]. The recovery time was 30 s between sets and one minute between exercises. The inclusion of sprinting during training sessions is considered a transfer exercise and has been included from the discipline analysis.

The plyometric program began with 48 contacts and ended with 144 contacts per session. The loaded plyometric group modified the standard plyometric routine by placing a load (2.5% of body mass) above the ankle joint. Maximal effort was encouraged verbally during all training sessions.

### 2.5. Testing Procedures

All field tests were performed on a tartan surface. Measurements were performed in a fixed order over four days, always preceded by a standardized warm-up. On the first test day, participants sprinted 40 m and then carried out two change-of-direction tests. On the second day, anthropometric measurements were followed by the two remaining change-of-direction tests. On the third day, the jump tests were followed by a Y-balance test, and on the final day the stork balance test and shuttle-run tests were performed. All tests were performed on a wooden surface. The warm-up for all tests included 5 min of submaximal running with change of direction exercises, 10 min of submaximal plyometrics (two jump exercises of 20 vertical (i.e., CMJ) and 10 horizontal jumps (i.e., two-footed ankle hop forward)), dynamic stretching exercises, and 5 min of a sprint-specific warm-up.

#### 2.5.1. Day 1

##### 40-m Sprint 

Subjects ran 40 m from a standing position, with times over 5 m, 10 m, 20 m, 30 m and 40 m recorded by paired photocells (Microgate, Bolzano, Italy). Three trials were separated by 6–8 min of recovery, with the fastest times noted for each distance.

##### Sprint 9–3–6–3–9 m with 180° Turns (S 180°)

During this test [21], subjects ran 9 m from the starting line to line A. Touching this line with one foot, they then made an 180° left- or right-hand turn. All subsequent turns were made in the same direction. They ran 3 m to line B, made another 180° turn, and ran 6 m forward. Then they made another 180° turn and ran a further 3 m forward, before making the final turn and running 9 m to the finish line (Figure 2).

##### Sprint 4 × 5 m (S4 × 5 m)

Five cones were set 5 m apart [21]. Subjects stood with their feet apart and a cone between their legs. At an acoustic signal, they ran 5 m to point A; there, they made a 90° turn to the right and ran 5 m to point B. After a second 90° turn, they ran to point C, where they made an 180° turn and ran back to the finish line (Figure 3).

#### 2.5.2. Day 2

##### Anthropometry

Anthropometric measurements included standing and sitting height (Holtain stadiometer, Crosswell, Crymych, GM, UK, accuracy 0.1 cm) and body mass (Tanita scales, BF683W, Sindelfingen, Germany, accuracy 0.1 kg). The overall percentage of body fat was estimated from the biceps, triceps, subscapular, and supra iliac skin-folds, using the equations of Durnin and Womersley for adolescent males aged 16.0-19.9 years [22]:% Body fat = [4.95/(Density − 4.5)] × 100(2)
where D = 1.162 − 0.063 (Log sum of four skinfolds).

##### Repeated Change of Direction Test

This test comprised 6 × 20 m sprints, each beginning from a standing position (Figure 4), 0.2 m behind the sensor, with 25-s active recovery intervals [23]. Times were measured using infrared sensors (Microgate, Bolzano, Italy) located 0.5 m above the ground at the start and finish lines. Four 100° turns were made at 4 m intervals. During the active recovery phase, subjects jogged slowly back to the starting line. The best time in a single trial (BT), the average time for the 6 × 20 m sprints (MT), and the total time for the six sprints (TT) were recorded, and the decrement (DEC) was calculated according to the formula [24]:DEC = 100 × (total sprint time ÷ ideal sprint) − 100(3)
where:Total sprint = sum of sprint times from all sprintsIdeal sprint = the number of sprints × best sprint time

##### Sprint 9–3–6–3–9 m with Backward and Forward Running (SBF)

Participants covered the same distance as in the S180° (Figure 2) test, but instead of making a turn, they shifted from forward to backward running. After the starting signal, they ran 9 m from the starting line to line B. Having touched line B with one foot, they ran 3 m backwards to line C and there changed to forward running. After 6 m, they ran 3 m backward and then ran the final 9 m forward to the finish line [21]. Each test was carried out three times, with a pause of 3 min between trials. The pause between the two tests was 7.5 min. The fastest values were recorded.

#### 2.5.3. Day 3

##### Vertical JUMP

After a 15-min warm-up, jump height was assessed using an infrared photocell mat connected to a digital computer (Opto-jump System, Microgate SARL, Bonzano, Italy). Contact and flight times were measured with a precision of 1 ms. The squat jump began at a knee angle of 90 degrees; avoiding any downward movement, subjects performed a vertical jump by pushing upwards, keeping their legs straight throughout. The counter-movement jump began from an upright position, with subjects making a rapid downward movement to a knee angle of 90° degrees and simultaneously beginning to push-off. During the third test, the subjects freely used their hands while jumping. One minute of rest was allowed between each of three trials, the highest values for each jump being used in subsequent analyses.

##### Five Jump Test

From an upright standing position, participants tried to cover as much distance as possible with five forward jumps by alternating left- and right-leg ground contacts. Distances were measured to the nearest 1 cm using a tape measure.

##### Dynamic Balance (Y-Balance Test)

The Y-balance protocol was similar to that described previously, and has a high reliability [13]. Reach directions were evaluated by affixing tape measures to the floor, one oriented anteriorly, and the other two running at 135° in the posterior-medial and posterior-lateral directions. All testing was conducted barefoot. Subjects stood on the dominant leg, with the most distal aspect of their great toe at the center of the grid. They then reached in the specified direction, while maintaining a single-limb stance. Tests were classified as invalid if the participants (1) did not touch the line with the reach foot while maintaining weight bearing on the stance leg, (2) lifted the stance foot from the center grid, (3) lost balance at any point during the trial, (4) did not maintain start and return positions for one full second, or (5) touched the reach foot down to gain support. The average maximum reach across the three directions (normalized for leg length) was calculated as a composite score for each subject [25]. After a demonstration, the participant completed four practice trials in each direction. Following a two-minute rest period, three definitive trials were made in each direction.

#### 2.5.4. Day 4

##### Static (Stork) Balance Test

On command, the subject raised the heel of one foot from the floor and placed it against the inside of the supporting knee, with both hands on the hips, maintaining balance for as long as possible. The trial ended if the participant moved his hands from his hips, if the ball of the dominant foot moved from its original position, or if the heel touched the floor. This test was carried out on the dominant leg, with the eyes open. The stopwatch-recorded score was the best of three attempts. Previous test–retest reliability scores with a similar adolescent population have been high (error of measurement 0.3 to 3.2%).

##### Yo-Yo Intermittent Recovery Test Level 1

A 20-m shuttle run was performed at increasing velocities to exhaustion, with 10 s of active recovery between runs. The test ended when the player twice failed to arrive within 2 m of the end line in the allotted time. The total distance (m) covered (including the last incomplete shuttle) was scored.

### 2.6. Statistical Analyses

Statistical analyses were carried out using the SPSS 23 program for Windows (SPSS, Inc., Armonk, NY, USA, IBM Corp). Means and SDs were calculated. Training-related effects were assessed by ANOVA (2-way analyses of variance (group × time)). If a significant F value was observed, Tukey’s post hoc procedure located pair-wise differences. Effect sizes were determined by converting partial eta-squared to Cohen’s d [26]; classified as small (0.00 ≤ d ≤ 0.49), medium (0.50 ≤ d ≤ 0.79), and large (d ≥ 0.80). Percentage changes were calculated as ([post-training value-pre-training value]/pre-training value) × 100, with the exception that for sprinting and the three change of direction tests percentage changes were calculated as ([pre-training value-post-training value]/pre-training value) × 100. The reliabilities of ability to change direction tests and sprint running performance measurements were assessed using intra-class correlation coefficients (ICC) [27]; all measurements of ability to change direction and sprinting reaching an acceptable reliability (r > 0.80) (Table 3). We accepted *p* ≤ 0.05 as our criterion of statistical significance.

## 3. Results

No pain, soreness or injury was observed during training. Test results before and after interventions are summarized in Table 3, Table 4 and Table 5 and Figure 5, Figure 6 and Figure 7. The loaded plyometric group developed significant shortening of all sprint times relative to the control and unloaded plyometric training groups, except over a distance of 40 m (where changes were similar for the two types of plyometrics). The unloaded plyometric program yielded increases relative to control over 5 and 30 m distances.

The loaded plyometrics enhanced ability to change direction on all three tests (Table 4; Figure 5), whereas no change of performance was seen in either the unloaded plyometrics or the control groups. Both forms of plyometric training enhanced jump performance relative to controls, with no differences between the two experimental groups (Figure 6). In terms of repeated change of direction (Table 5; Figure 7), best times, decrements and total times were largely enhanced in both experimental groups relative to the controls, but the loaded plyometric training also achieved shorter mean times relative to unloaded plyometrics. In contrast, the shuttle run scores remained unchanged for all groups. The stork balance scores increased with loaded plyometrics, but the unloaded plyometric training did not enhance static balance relative to controls (Table 6). Finally, the Y-balance scores remained unchanged for all three groups.

## 4. Discussion

The aim of the present study was to compare the effects of plyometric training with and without external loading on sprinting, ability to change direction, jumping, to make repeated changes of direction, maximal aerobic power and balance of elite U-17 male soccer players at a critical phase in their playing season. On most measures, gains were similar for both experimental groups, but the loaded plyometrics yielded larger gains than unloaded plyometrics for sprinting over short distances, rapid changes of direction and static balance, all qualities useful to the soccer player.

Several previous studies have shown the effectiveness of plyometric training in improving sprint performance [8,9,13], and our present observations suggest that this response can be increased by external loading, probably because greater strength was developed by this tactic; it also seems important to introduce horizontal acceleration (skipping and jumping with horizontal displacements) [28] in order to increase sprint speeds.

Contrary to the gains in change of direction ability noted for all three tests in this study, Hammami et al. 2016 [13] found no significant change in this skill among U17 soccer players after 8 weeks of plyometric training alone, pointing to the likely value of the external loading that was used here. The rate of force development is a pre-eminent factor when changing direction [29], and the eccentric strength of the thigh muscles is also critical to the deceleration phase of impulsive movement when changing direction [30,31,32]. Decreased ground reaction times may also increase muscular force output and movement efficiency, positively affecting ability to change direction [29].

Previous authors have shown that combinations of plyometrics with various types of strength training increase vertical jump performance [8,9,13]. The present findings demonstrated higher jump performance in both experimental groups after training. This might be explained by an enhanced motor neuron excitability and neuromuscular adaptations [5], or better use of the stretch-shortening cycle [33].

Like us, Negra et al. 2017 [7] reported an improvement of stork balance test scores when prepubertal male soccer players undertook 8 weeks of plyometric training on a combination of stable and unstable surfaces. This response may be related to either an improved co-contraction of the lower extremity muscles [34] or changes in proprioception and neuromuscular control [6].

The current results found no improvements in predicted maximal aerobic power following the intervention. Likewise, Michailidis et al. 2019 [6] found no significant change in shuttle-run scores when youth soccer players undertook 6 weeks of a combination of soccer training, plyometric training and change of direction exercises, and De Villareal et al. 2015 [35] saw no gains of shuttle-run performance with a combination of plyometric and sprint training. However, Ramirez-Campillo et al. 2020 [8] observed increases of 20-m multistage shuttle running speed when young soccer players undertook plyometric training before or after soccer practice. Where improvements have been observed, these may reflect enhanced running efficiency [36] or an increase of tendon stiffness [5] that allows a faster transfer of force from the contracting muscles, reducing reaction times and improving the ability to change direction [5,36].

The apparent superiority of the externally loaded training probably reflects the overload principle, with the muscles showing a greater adaptive response when stressed beyond their normal capacity [37]. It may be speculated that the external loads enabled players to apply greater amounts of force against the ground over a longer time, generating higher impulses during the jumps [16], and thus facilitating greater adaptations.

### Practical Applications

This study underlines that ten weeks of plyometric training with external loading enhances several attributes important to soccer performance to an extent greater than the allocation of a similar time to plyometric training alone. Performance in soccer relies greatly on the strength and power of the lower limbs, and as this study has demonstrated it is practicable to incorporate a substantial volume of plyometric training with external loading into traditional in-season technical and tactical training sessions. Such initiatives induce substantial gains in several performance measures important to the playing potential of soccer players.

## 5. Conclusions

We conclude that the introduction of 10 weeks of in-season loaded plyometrics into the regimen of U17 male soccer players yields gains in several physical performance test scores relative to either unloaded plyometrics or the control training regimen.

## Figures and Tables

**Figure 1 ijerph-17-07877-f001:**
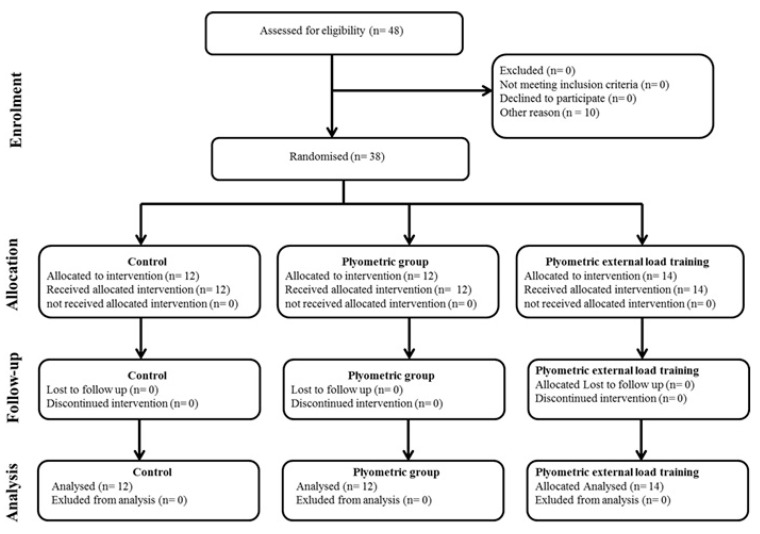
The diagram (The CONSORT: Consolidated Standards of Reporting Trials) includes detailed information on the interventions received.

**Figure 2 ijerph-17-07877-f002:**
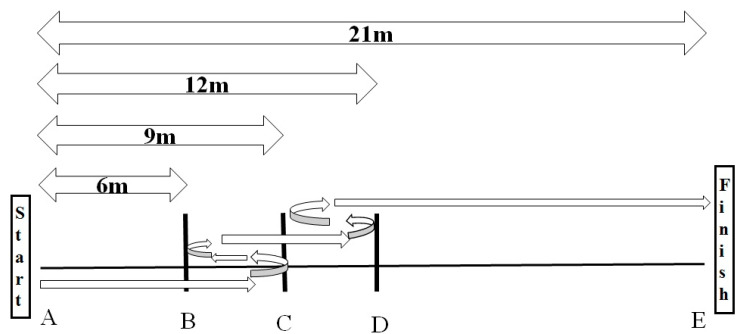
Design of the sprint 9–3–6–3–9 m with 180° turns (S180°) or with backward and forward running (SBF). A and E are start and finish lines respectively. B, C and D are lines of change of direction.

**Figure 3 ijerph-17-07877-f003:**
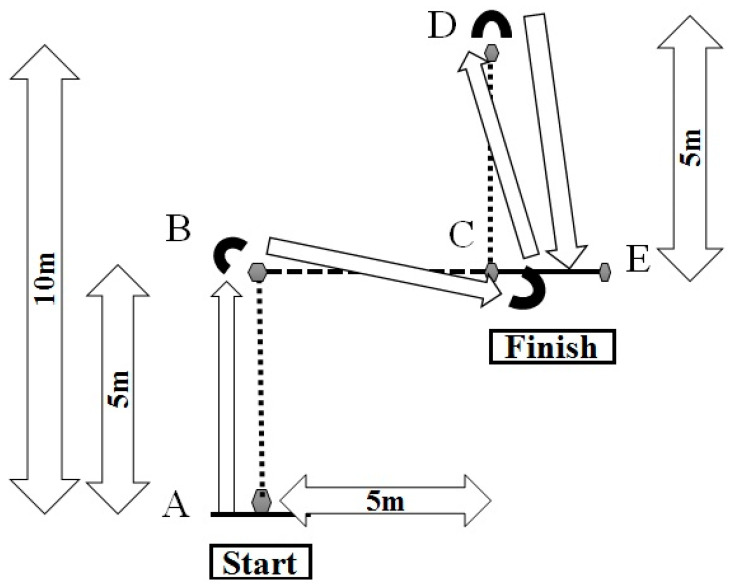
Design of the sprint 4 × 5 m (S4 × 5 m). A and E are start and finish lines respectively. B, C and D are lines of change of direction.

**Figure 4 ijerph-17-07877-f004:**
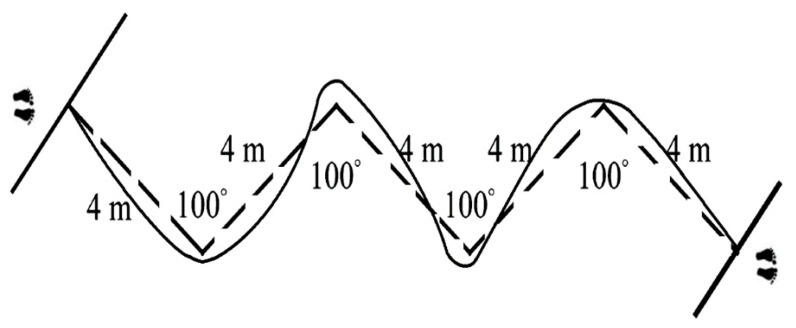
Design of the repeated change-of-direction (RCOD) test.

**Figure 5 ijerph-17-07877-f005:**
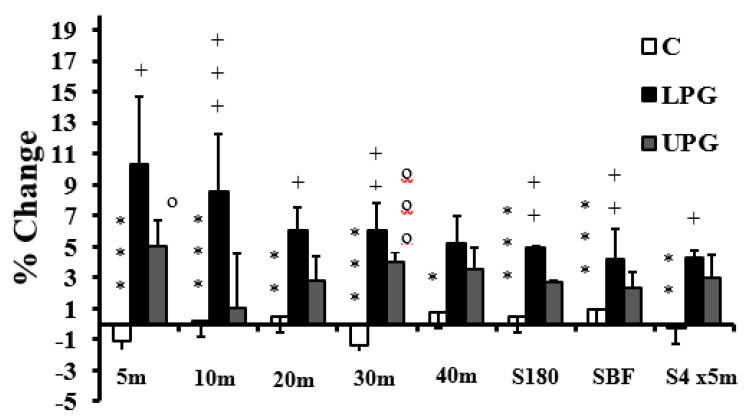
Training associated changes in sprint performance and ability to change direction in loaded plyometric group (LPG), unloaded plyometric group (UPG) and controls (C). S180° = sprint time over 9–3–6–3–9 m with 180° turns; SBF = sprint time over 9–3–6–3–9 m with backward and forward running; S4 × 5 = sprint time over 4 × 5 m; * = denotes a significant difference between LPG and C; + = denotes a significant difference between LPG and PG; o = denotes a significant difference between UPG and C; *: *p* < 0.05; **: *p* < 0.01; ***: *p* < 0.001; +: *p* < 0.05; ++: *p* < 0.01; +++: *p* < 0.001; o: *p* < 0.05; ooo: *p* < 0.001.

**Figure 6 ijerph-17-07877-f006:**
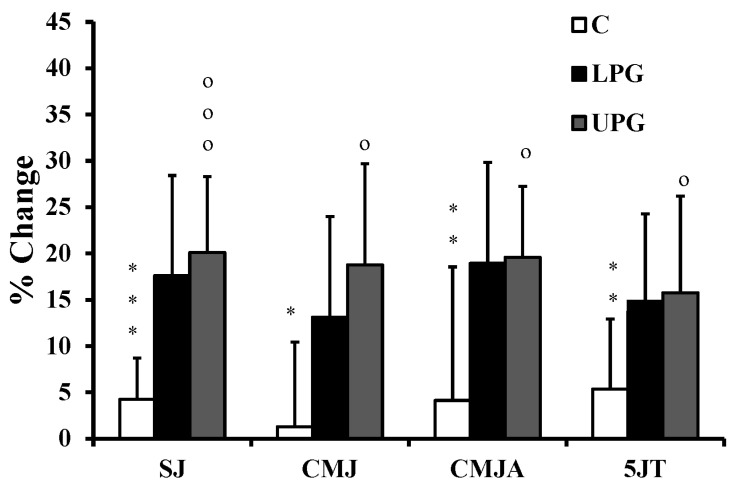
Training associated changes in vertical (SJ: Squat Jump, CMJ: countermovement jump: CMJ; CMJA: countermovement with aimed arms) and horizontal jumps (5JT: Five jump test) in loaded plyometric group (LPG), unloaded plyometric group (UPG) and controls (C). * = denotes a significant difference between LPG and C; + = denotes a significant difference between LPG and UPG; o = denotes a significant difference between UPG and C; *: *p* < 0.05; **: *p* < 0.01; ***: *p* < 0.001; o: *p* < 0.05; ooo: *p* < 0.001.

**Figure 7 ijerph-17-07877-f007:**
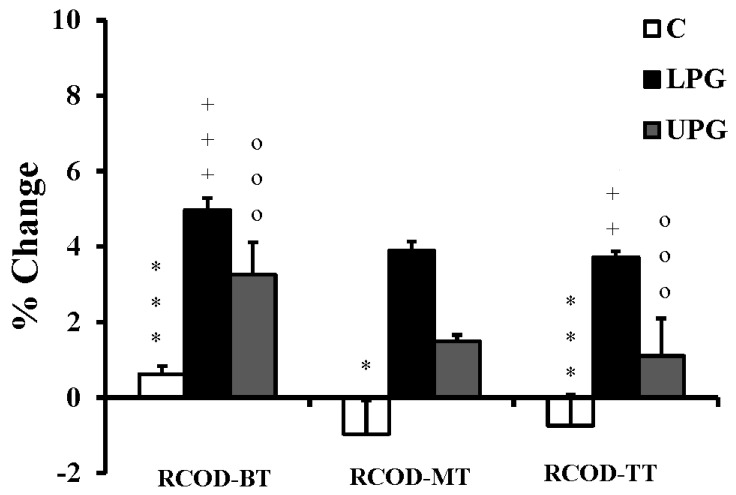
Training associated changes in Repeated Change of Direction (RCOD) test parameters (BT: best time; MT: Mean time; TT: Total time) in loaded plyometric group (LPG), unloaded plyometric group (UPG) and controls (C). * = denotes a significant difference between LPG and C; + = denotes a significant difference between LPG and UPG; o = denotes a significant difference between UPG and C; *: *p* < 0.05; ***: *p* < 0.001; ++: *p* < 0.01; +++: *p* < 0.001; ooo: *p* < 0.001.

**Table 1 ijerph-17-07877-t001:** The used PEDro scale for reporting the three randomized divided groups.

1	Eligibility criteria	yes
2	Randomized allocation	yes
3	Concealed allocation	yes
4	Comparable at baseline	yes
5	Blinded subjects	no
6	Blinded therapist	yes
7	Blinded assessors	no
8	Adequate follow up	yes
9	Intention to treat analysis	no
10	Between group comparisons	yes
11	Point ability estimates and variance	yes
Total Score		7/10

**Table 2 ijerph-17-07877-t002:** Details of two types of plyometric training.

	Weeks 1 and 2	Weeks 3 and 4	Weeks 5 and 6	Weeks 7 and 8	Weeks 9 and 10
Exercise 1: 3 hops to the right then 3 hops to the left (6 ground contacts), finally sprint 20 m.	2 Repetitions	3 Repetitions	4 Repetitions	5 Repetitions	6 Repetitions
Exercise 2: 6 lateral 0.3 m hurdle jumps (3 to left and 3 to right) (6 ground contacts), then sprint 20 m.	2 Repetitions	3 Repetitions	4 Repetitions	5 Repetitions	6 Repetitions
Exercise 3: 6 horizontal jumps (three bell feet horizontal with the right leg follows from three bell feet horizontal with the left leg) (6 ground contacts), then sprint 20 m.	2 Repetitions	3 Repetitions	4 Repetitions	5 Repetitions	6 Repetitions
Exercise 4: 6 × 0.4 m hurdle jumps (6 ground contacts), then sprint 20 m.	2 Repetitions	3 Repetitions	4 Repetitions	5 Repetitions	6 Repetitions
Total jump per each session	48 (36 unilateral jump + 12 bilateral jump)	72 (54 unilateral jump + 18 bilateral jump)	96 (72 unilateral jump + 24 bilateral jump)	120 (90 unilateral jump + 30 bilateral jump)	144 (98 unilateral jump + 36 bilateral jump)

**Table 3 ijerph-17-07877-t003:** Inter-class correlation coefficient (ICC) and coefficient of variation (CV), showing acceptable reliability for measures of track running velocity, change of direction, jump tests, static and dynamic balance tests.

Parameters	ICC	CV
Sprint		
5 m	0.90	1.8
10 m	0.87	0.9
20 m	0.89	1.8
30 m	0.85	1.4
40 m	0.90	1.6
Change of direction		
Sprint time over 9–3–6–3–9 m with 180° turns	0.91	1.8
Sprint time over 9–3–6–3–9 m with backward and forward running	0.89	1.5
Sprint time over 4 × 5 m	0.89	2.1
Jump tests		
Squat jump	0.86	5.1
Countermovement jump	0.87	5.0
Countermovement jump aimed arms	0.86	4.6
Five jump test	0.87	5.5
Y Balance Test		
Right support leg		
Right leg/Left	0.98	9.5
Right leg/Back	0.97	7.5
Right leg/Right	0.98	19.2
Left support leg		
Left leg/Left	0.98	10.3
Left leg/Back	0.98	7.6
Left leg/Right	0.97	18.2
Stork Balance Test		
Right leg	0.76	69.5
Left leg	0.88	70.6

**Table 4 ijerph-17-07877-t004:** Inter-group comparison of sprint running, change of direction and jump performances before (pre) and after (post) the 10-week trial.

Variables		Group	Pre-Trial	Post-Trial	*p* Value	d (Cohen)
Sprint	5 m (s)	Loaded plyometric group	1.17 ± 0.02	1.05 ± 0.06 ***^,+^	<0.001 a	1.36 (large)
		Unloaded plyometric group	1.16 ± 0.02	1.12 ± 0.05 ^o^	<0.001 b	1.26 (large)
		Control	1.17 ± 0.02	1.18 ± 0.05	<0.001 c	1.42 (large)
	10 m (s)	Loaded plyometric group	2.12 ± 0.02	1.94 ± 0.08 ***^,+++^	<0.001 a	1.51 (large)
		Unloaded plyometric group	2.12 ± 0.02	2.10 ± 0.08	<0.001 b	1.34 (large)
		Control	2.12 ± 0.02	2.12 ± 0.06	<0.001 c	1.58 (large)
	20 m (s)	Loaded plyometric group	3.26 ± 0.05	3.07 ± 0.09 **^,+^	0.003 a	0.85 (large)
		Unloaded plyometric group	3.27 ± 0.06	3.08 ± 0.1	<0.001 b	1.04 (large)
		Control	3.24 ± 0.07	3.25 ± 0.08	<0.001 c	1.17 (large)
	30 m (s)	Loaded plyometric group	4.55 ± 0.06	4.27 ± 0.08 ***^,++^	<0.001 a	1.74 (large)
		Unloaded plyometric group	4.56 ± 0.05	4.42 ± 0.15 ^ooo^	<0.001 b	1.29 (large)
		Control	4.53 ± 0.07	4.62 ± 0.12	<0.001 c	1.5 (large)
	40 m (s)	Loaded plyometric group	5.85 ± 0.10	5.55 ± 0.13 *	0.021 a	0.68 (medium)
		Unloaded plyometric group	5.87 ± 0.09	5.68 ± 0.15	<0.001 b	1.42 (large)
		Control	5.85 ± 0.08	5.79 ± 0.24	0.027 c	0.65 (medium)
Change of direction	Sprint time over 9–3–6–3–9 m with 180° turns (s)	Loaded plyometric group	8.73 ± 0.14	8.30 ± 0.14 ***^,++^	<0.001 a	1.19 (large)
		Unloaded plyometric group	8.78 ± 0.14	8.54 ± 0.14	<0.001 b	1.62 (large)
		Control	8.71 ± 0.19	8.70 ± 0.13	<0.001 c	1.22 (large)
	Sprint time over 9–3–6–3–9 m with backward and forward running (s)	Loaded plyometric group	8.62 ± 0.14	8.26 ± 0.17 ***^,++^	<0.001 a	1.06 (large)
		Unloaded plyometric group	8.67 ± 0.14	8.47 ± 0.14	<0.001 b	1.44 (large)
		Control	8.65 ± 0.13	8.57 ± 0.13	0.004 c	0.82 (large)
	Sprint time over 4 × 5 m (s)	Loaded plyometric group	6.33 ± 0.16	6.08 ± 0.18 **^,+^	0.003 a	0.85 (large)
		Unloaded plyometric group	6.38 ± 0.14	6.26 ± 0.23	0.004 b	0.71 (medium)
		Control	6.36 ± 0.13	6.37 ± 0.20	0.026 c	0.66 (medium)
Jump tests	Squat jump (cm)	Loaded plyometric group	33.7 ± 1.7	39.6 ± 2.8 ***	<0.001 a	1.03 (large)
		Unloaded plyometric group	33.4 ± 1.9	40.0 ± 2.8 ^ooo^	<0.001 b	2.18 (large)
		Control	33.6 ± 1.7	35.0 ± 2.0	<0.001 c	1.06 (large)
	Contermovement jump (cm)	Loaded plyometric group	35.5 ± 1.7	40.2 ± 4.4 *	0.010 a	0.75 (medium)
		Unloaded plyometric group	35.2 ± 1.9	41.7 ± 3.4 ^o^	<0.001 b	1.37 (large)
		Control	35.7 ± 1.8	36.1 ± 3.2	0.002 c	0.88 (large)
	Contermovement jump aimed arms (cm)	Loaded plyometric group	37.3 ± 1.7	44.3 ± 2.9 **	0.005 a	0.80 (large)
		Unloaded plyometric group	36.7 ± 1.6	43.9 ± 3.7 ^o^	<0.001 b	1.87 (large)
		Control	37.5 ± 1.8	38.9 ± 4.5	0.001 c	0.94 (large)
	Five jump test (m)	Loaded plyometric group	11.5 ± 0.8	13.2 ± 0.7 **	0.005 a	0.81 (large)
		Unloaded plyometric group	11.4 ± 0.5	13.0 ± 0.8 ^o^	<0.001 b	1.82 (large)
		Control	11.4 ± 0.6	12.0 ± 0.9	0.016 c	0.71 (large)

s = seconds; * = denotes a significant difference between loaded and control groups; + = denotes a significant difference between loaded and unloaded groups; o = denotes a significant difference between unloaded and control groups; a = denotes a main effect of group, b = denotes a main effect of time; c = denote a group × time interaction. *: *p* < 0.05; **: *p* < 0.01; ***: *p* < 0.001; ^+^: *p* < 0.05; ^++^: *p* < 0.01; ^+++^: *p* < 0.001; ^o^: *p* < 0.05; ^ooo^: *p* < 0.001.

**Table 5 ijerph-17-07877-t005:** Comparison of repeated change of direction and YYIRTL1 performance between groups before (pre) and after (post) the 10-week trial.

Variables		Group	Pre-Trial	Post-Trial	*p* Value	d (Cohen)
Repeated change of direction test	Repeated change of direction-BT (s)	Loaded plyometric group	6.41 ± 0.07	6.07 ± 0.07 ***^,+++^	<0.001 a	2.14 (large)
		Unloaded plyometric group	6.42 ± 0.04	6.21 ± 0.07 ^ooo^	<0.001 b	3.33 (large)
		Control	6.41 ± 0.05	6.37 ± 0.05	<0.001 c	2.09 (large)
	Repeated change of direction-MT (s)	Loaded plyometric group	6.60 ± 0.07	6.13 ± 0.80 *	0.014 a	0.72 (medium)
		Unloaded plyometric group	6.60 ± 0.07	6.50 ± 0.07	0.041 b	0.50 (medium)
		Control	6.62 ± 0.05	6.68 ± 0.07	0.025 c	0.66 (medium)
	Repeated change of direction-DEC (%)	Loaded plyometric group	4.45 ± 0.95	4.17 ± 0.45 **	0.001 a	0.93 (large)
		Unloaded plyometric group	4.23 ± 0.07	3.99 ± 1.36 ^ooo^	0.651 b	0.10 (small)
		Control	4.67 ± 1.42	5.47 ± 0.05	0.065 c	0.56 (medium)
	Repeated change of direction-TT (s)	Loaded plyometric group	39.60 ± 0.45	38.13 ± 0.41 ***^,++^	<0.001 a	1.94 (large)
		Unloaded plyometric group	39.51 ± 0.48	39.07 ± 0.50 ^ooo^	<0.001 b	1.22 (large)
		Control	39.66 ± 0.41	40.03 ± 0.34	<0.001 c	1.82 (large)
Yo-Yo intermittent recovery test level 1	Aerobic maximum speed (km.h^−1^)	Loaded plyometric group	16.5 ± 0.5	17.0 ± 0.4	0.054 a	0.58 (medium)
		Unloaded plyometric group	16.4 ± 0.4	17.0 ± 0.7	<0.001 b	0.89 (large)
		Control	16.3 ± 0.7	16.5 ± 0.5	0.609 c	0.23 (small)
	Total distance covered (m)	Loaded plyometric group	1851 ± 247	2197 ± 348	0.01 a	0.75 (medium)
		Unloaded plyometric group	1950 ± 164	2133 ± 522	0.004 b	0.71(medium)
		Control	1630 ± 427	1860 ± 415	0.715 c	0.20 b(small)

BT = best time; MT = mean time; DEC = decrement; TT = total time; s = second; YYIRTL1=Yo-Yo intermittent recovery test level 1; * = denotes a significant difference between loaded and control groups; + = denotes a significant difference between loaded and unloaded groups; o = denotes a significant difference between unloaded and control groups; a = denotes a main effect of group, b = denotes a main effect of time; c = denote a group × time interaction. *: *p* < 0.05; **: *p* < 0.01; ***: *p* < 0.001; ^++^: *p* < 0.01; ^+++^: *p* < 0.001; ^ooo^: *p* < 0.001.

**Table 6 ijerph-17-07877-t006:** Comparison of balance performance between groups before (pre) and after (post) the 10-week trial.

Variables			Group	Pre-Trial	Post-Trial	*p* Value	d (Cohen)
Y Balance Test	Right support leg	Right leg/Left (cm)	Loaded plyometric group	86 ± 8	91 ± 8 *	0.027 a	0.65 (medium)
			Unloaded plyometric group	82 ± 8	86 ± 7	0.097 b	0.40 (small)
			Control	83 ± 8	83 ± 6	0.471 c	0.29 (small)
		Right leg/Back (cm)	Loaded plyometric group	111 ± 7	118 ± 5 ^+^	0.014 a	0.71 (medium)
			Unloaded plyometric group	106 ± 10	110 ± 9	0.015 b	0.59 (medium)
			Control	109 ± 8	112 ± 7	0.679 c	0.21 (small)
		Right leg/Right (cm)	Loaded plyometric group	55 ± 9	60 ± 9	0.185 a	0.44 (small)
			Unloaded plyometric group	53 ± 12	53 ± 12	0.522 b	0.15 (small)
			Control	52 ± 10	52 ± 10	0.554 c	0.26 (small)
	Left support leg	Left leg/Left (cm)	Loaded plyometric group	88 ± 10	93 ± 10 ^+^	0.011 a	0.74 (medium)
			Unloaded plyometric group	82 ± 8	85 ± 5	0.066 b	0.44 (small)
			Control	84 ± 7	87 ± 7	0.873 c	0.12 (small)
		Left leg/Back (cm)	Loaded plyometric group	114 ± 6	121 ± 5 ***	0.001 a	0.98 (large)
			Unloaded plyometric group	109 ± 12	108 ± 10	0.286 b	0.25 (small)
			Control	113 ± 6	112 ± 7	0.155 c	0.46 (small)
		Left leg/Right (cm)	Loaded plyometric group	53 ± 8	55 ± 8	0.196 a	0.43 (small)
			Unloaded plyometric group	55 ± 11	53 ± 11	0.630 b	0.10 (small)
			Control	51 ± 11	47 ± 11	0.435 c	0.31 (small)
Stork Balance Test		Right leg (s)	Loaded plyometric group	2.24 ± 0.76	7.78 ± 4.50 *	0.022 a	0.68 (medium)
			Unloaded plyometric group	3.37 ± 2.94	4.86 ± 3.42	<0.001 b	1.00 (large)
			Control	2.16 ± 0.61	3.35 ± 2.53	0.011 c	0.74 (medium)
		Left leg (s)	Loaded plyometric group	2.08 ± 0.39	10.04 ± 6.79 **	0.008 a	0.76 (medium)
			Unloaded plyometric group	2.99 ± 2.71	6.91 ± 5.74	<0.001 b	1.13 (large)
			Control	1.78 ± 0.51	3.18 ± 3.13	0.017 c	0.70 (medium)

s = second; * = denotes a significant difference between loaded and control groups; + = denotes a significant difference between loaded and unloaded groups; a = denotes a main effect of group, b = denotes a main effect of time; c = denote a group x time interaction; *: *p* < 0.05; **: *p* < 0.01; ***: *p* < 0.001; ^+^: *p* < 0.05.

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
