# Peer review of "Effects of Unloaded vs. Ankle-Loaded Plyometric Training on the Physical Fitness of U-17 Male Soccer Players"

_ijerph, 2020, doi:10.3390/ijerph17217877_

Round 1

Reviewer 1 Report

Effects of unloaded vs. ankle-loaded plyometric training on the physical fitness of U-17 male soccer players.

The article reports an interesting analysis of a plyometric training program in junior male soccer players. Authors use several tests as evaluation of the efficiency of the plyometric exercises and compared to a control group.

The results display physical fitness enhancement of the players that carried out the training program. They improve in several physical skills related to football with a statistical significance compared to the control group.

Major comments

Authors claim that load plyometrics produces increases performance in football players more than unloaded plyometric training. But they clarify the load the added in the point 2.4 of the methods, which is hard to find, and it is important that be clarify from the beginning for readers.

Plyometric

Authors describes details such us the devices used in the assessments, the software used for statistical analysis, but do not describes what is a plyometric movement. What is the difference regarding to contractions with not plyometric regime?

Sprints are not plyometric exercise because there is not an accumulation of elastic energy, the movement is not an elastic-reflex movement. To be considered as a plyometric exercise, there must be a reactive-ballistic regime (Verkhoshansky 1961, 1970).

Several key authors in the plyometric field are missing, such as: Verkhoshansky, Siff, Matveev, Platonov, Zatsiorski, García Manso, etc.

The exercises used and described in the training program must reflect the plyometric component of it. The jumps exercises must point out that players performed all jumps in a row (in the same set), in a deformation/accumulation cycle.

Minor comments

Abstract

For general readers sprints 9-3-6-3-9 m with 180° turns (S180°) could be hard to understand.

10-weeks of loaded than unloaded plyometric training. Training time should be pointed out at the beginning in the text. Where the sample and method are described.

Keywords I recommend to use different keywords that do not appear in the title and the abstract

A new trend in plyometric training is the addition of external loading; This is not new. Please revise the literature about it.

Few previous studies have examined the impact of combined forms of plyometrics on 50 soccer players, What studies?

tests were repeated after completing the 10-week intervention, 5-9 days after the last training session. […]

[…] All participants were engaged in soccer training 6-7 times per week, played one official game per week from the beginning of the competitive season (September) until the end of the trial (March), and also had a weekly two hours of physical education course.

Why authors do not consider the effect of the training in season.

Table 2. Details of two types of plyometric training.

I recommend to use the word task or exercise, or even microcycle or session, instead of workshop

Why authors used a bioimpedance and skin-fold technique?

Why do not used the Yuhasz (1974) or Alan Martin Formula:  Anthropometric estimation of muscle mass in men. Medicine & Science in Sports & Exercise. 22(5):729-733, October 1990.MARTIN, A D; SPENST, L F; DRINKWATER, D T; CLARYS, J P

Vertical Jump. After a 15-min warm-up What was the warm up?

Why authors do not carried out an ANOVA ?

Instead of using currencies, why authors do not use other classic symbols used in data visualization, such us *, +, o, etc.?

Conclusions

Authors conclude that:

We conclude that the introduction of 10 weeks of in-season loaded plyometrics into the regimen of U17 male soccer players yields gains in several measures of performance relative to either unloaded plyometrics or the control training regimen.

But in the abstract, we can read:

We conclude that measures of physical fitness (sprint, change of direction, repeated 30 change of direction, and dynamic balance) in male soccer players were increased more by 10-weeks 31 of loaded than unloaded plyometric training.

The physical fitness improves, but measures of performance in football is much more than the physical condition of the players. Therefore, this can lead to a misunderstanding, because the conclusions should only refer to the physical assessment of the players and their results, not the performance in football.

Author Response

Responses to reviewer-1 are highlighted in Yellow

The article reports an interesting analysis of a plyometric training program in junior male soccer players. Authors use several tests as evaluation of the efficiency of the plyometric exercises and compared to a control group.

The results display physical fitness enhancement of the players that carried out the training program. They improve in several physical skills related to football with a statistical significance compared to the control group.

Author responses:

Thank you for your support to our study and your positive report.

Major comments

 Reviewer question

Authors claim that load plyometrics produces increases performance in football players more than unloaded plyometric training. But they clarify the load the added in the point 2.4 of the methods, which is hard to find, and it is important that be clarify from the beginning for readers.

Author responses:

Thank you for your pertinent remark.

We added sentences (2.5% of body mass placing in the ankle joint in the Abstract, Introduction and in methods.

Reviewer question

Plyometric

Authors describes details such us the devices used in the assessments, the software used for statistical analysis, but do not describes what is a plyometric movement. What is the difference regarding to contractions with not plyometric regime?

Author responses:

Thank you for your pertinent remark.

We added more information about Plyometrics in the Introduction section.

“Plyometric training is an exercise in which an eccentric muscle contraction is quickly followed by a concentric muscle contraction. In other words, when a muscle is rapidly contracted and lengthened, and then immediately followed with a further contraction and shortening, this is a plyometric exercise [10-12]. These method of training is a type of strength training and it is very popular applied in many sports such as soccer, handball, basketball [5, 13, 14]. It involves performing bodyweight jumping-type exercises and throwing medicine balls using the so-called stretch-shortening cycle (SSC) muscle action [5]. The SSC enhances the ability of the neural and musculotendinous systems to produce maximal force in the shortest amount of time, prompting the use of plyometric exercise as a bridge between strength and speed [15]. In this regard, plyometric exercises has been extensively used for augmenting dynamic athletic performance, such as sprint, change of direction and jump performance [5, 13, 14]. The plyometric training with additional external load will become from day to day discussed by several scientific articles.”

Reviewer question

Sprints are not plyometric exercise because there is not an accumulation of elastic energy, the movement is not an elastic-reflex movement. To be considered as a plyometric exercise, there must be a reactive-ballistic regime (Verkhoshansky 1961, 1970).

Author responses:

Thank you for your pertinent remak

We are agree with you. The inclusion of sprinting during training sessions is considered a transfer exercise and has expired from discipline analysis.

Reviewer question

Several key authors in the plyometric field are missing, such as: Verkhoshansky, Siff, Matveev, Platonov, Zatsiorski, García Manso, etc.

 Author responses:

Thank you for your pertinent remark.

We added most of the suggested references in different section of the article.

Reviewer question

The exercises used and described in the training program must reflect the plyometric component of it. The jumps exercises must point out that players performed all jumps in a row (in the same set), in a deformation/accumulation cycle.

 Author responses:

Thank you for your remark. The modification was done in “Details of standard and modified plyometric training” section: “Each exercise set comprised 6 jumps (6 contacts) and a sprint of 20 meters [14].

Minor comments

Reviewer question

Abstract

For general readers sprints 9-3-6-3-9 m with 180° turns (S180°) could be hard to understand.

Author responses:

Thank you for your pertinent remark.

We agree with you, but it is the name of the test proposed previously in the scientific literature (Sporis, G,. et al. (2010). We kindly ask reviewer to keep this same name for this test.

Sporis, G, Jukic, I, Milanovic, L, and Vucetic, V. Reliability and factorial validity of agility tests for soccer players. J Strength Cond Res 24: 679–686, 2010.

Reviewer question

 10-weeks of loaded than unloaded plyometric training. Training time should be pointed out at the beginning in the text. Where the sample and method are described.

Author responses:

Thank you for your pertinent remark. However, as mentioned in “Participants” section, in the current investigation a team of experienced elite male soccer players was assigned randomly between three groups: unloaded plyometric training (UP; n = 12), loaded plyometric training (LP; n = 14) and controls (C; n = 12). The two experimental group contained different soccer players and they performed the plyometric training program during the same period of the season.

 Reviewer question

Keywords I recommend to use different keywords that do not appear in the title and the abstract

Author responses:

Thank you for your pertinent remark. We changed the “Keywords” section.

 Reviewer question

A new trend in plyometric training is the addition of external loading; This is not new. Please revise the literature about it.

Author responses:

Ok, We changed the sentence.

 Reviewer question

Few previous studies have examined the impact of combined forms of plyometrics on 50 soccer players, What studies?

Author responses:

Ok, we added references.

 Reviewer question

tests were repeated after completing the 10-week intervention, 5-9 days after the last training session. […]

Author responses:

Authors do not understand this question from the reviewer. We kindly ask the reviewer to clarify his/her query if the article still lacks clarity

Reviewer question

[…] All participants were engaged in soccer training 6-7 times per week, played one official game per week from the beginning of the competitive season (September) until the end of the trial (March), and also had a weekly two hours of physical education course.

Author responses:

Authors do not understand the question of the reviewer. We kindly ask the reviewer to clarify his/her query if the article still lacks clarity .

 Reviewer question

Why authors do not consider the effect of the training in season.

Author responses:

Thank you for your pertinent remark.

Unfortunately, we did not study the effect of training throughout the season because there are players who belong to the national team, and therefore there is a lot of training for the national team and subsequently we cannot rectify and control the training program and load.

 Reviewer question

Table 2. Details of two types of plyometric training.

I recommend to use the word task or exercise, or even microcycle or session, instead of workshop

Author responses:

Thank you for your pertinent remark. We changed workshop to exercise in all the text.

 Reviewer question

Why authors used a bioimpedance and skin-fold technique?

Why do not used the Yuhasz (1974) or Alan Martin Formula:  Anthropometric estimation of muscle mass in men. Medicine & Science in Sports & Exercise. 22(5):729-733, October 1990.MARTIN, A D; SPENST, L F; DRINKWATER, D T; CLARYS, J P

Author responses:

Thank you for your pertinent remark.

We used the bioimpedance and skin-fold technique because this technique is used in scientific research articles. Moreover, it is a valid and easy method to estimate %body fat. In addition, the study of Durnin and Womersly developed an equation to different age and sex of athletes (young vs adults; male vs female).

Reviewer question

Vertical Jump. After a 15-min warm-up What was the warm up?

Author responses:

Ok, we added detail of warm-up in the testing procedures section.

“All tests were performed on a wooden surface at the same time of day. The warm-up for all tests included 5 min of submaximal running with change of direction exercises, 10 minutes of submaximal plyometrics (2 jump exercises of 20 vertical [i.e., CMJ] and 10 horizontal jumps [i.e., 2-footed ankles hop forward]), dynamic stretching exercises, and 5 minutes of a sprint-specific warm-up.””

 Reviewer question

Why authors do not carried out an ANOVA ?

Author responses:

Thank you for your pertinent remark. We used ANOVA.

We added “ ANOVA” in the statistical analyses section.

 Reviewer question

Instead of using currencies, why authors do not use other classic symbols used in data visualization, such us *, +, o, etc.?

Author responses:

We added the symbols that you proposed.

 Reviewer question

Conclusions

Authors conclude that:

We conclude that the introduction of 10 weeks of in-season loaded plyometrics into the regimen of U17 male soccer players yields gains in several measures of performance relative to either unloaded plyometrics or the control training regimen.

But in the abstract, we can read:

We conclude that measures of physical fitness (sprint, change of direction, repeated change of direction, and dynamic balance) in male soccer players were increased more by 10-weeks of loaded than unloaded plyometric training.

Author responses:

Thank you for your pertinent remark. Ok, we changed in the “Abstract” section.

 Reviewer question

The physical fitness improves, but measures of performance in football is much more than the physical condition of the players. Therefore, this can lead to a misunderstanding, because the conclusions should only refer to the physical assessment of the players and their results, not the performance in football.

Author responses:

Thank you for your pertinent remark.

We changed « measures performance » by « physical assessment ».

Reviewer 2 Report

Nice study, well design and with very interesting results. Congratulations

Strength:

The research area it's innovate and has a lot of potential and it's practical and applied;

A very interesting study design, sample, methods and also a little luke "No pain, soreness or injury was observed during training"

Results and discussion are very direct, but well explained

Weakness:

In this part, I can give my opinion and here I can say that the introduction could be a bitter lounger and more deeply;

This measures body mass; height and body fat should be presented with the results before and after the training programs;

They should present the position of the players in the pitch (goalkeeper, defender, etc);

They should present the time of play that each group had, because it's 90' per week, about 900' in total;

The tests should be taken in a grass surface instead of a tartan surface.

In opinion it's a very good article, well written, interesting, very applied and with a lot of interest for both researchers and coaches.

Author Response

Responses to reviewer-2 are highlighted in Green

Comments and Suggestions for Authors

Nice study, well design and with very interesting results. Congratulations

Strength:

The research area it's innovate and has a lot of potential and it's practical and applied;

A very interesting study design, sample, methods and also a little luke "No pain, soreness or injury was observed during training"

Results and discussion are very direct, but well explained

Weakness:

Reviewer question

In this part, I can give my opinion and here I can say that the introduction could be a bitter lounger and more deeply;

Author responses:

Thank you for your pertinent remark.

This question was asked byreviewer-1 and we have answered (we have deepened the introduction).

Reviewer question

This measures body mass; height and body fat should be presented with the results before and after the training programs;

Author responses:

We are agreeing with you, but we don’t make anthropometric measures before and after the training because:

  • The training intervention was not for a long period; it is short one (only 10 weeks). Thereafter, we can suppose that such short period could not lead to great changes in anthropometrics variables such as height.
  • We did not assess the %body fat and body mass because the aim of the current study was to investigate the effect of plyometric training on physical performance not in anthropometric parameters. However, this could be considered as a limitation of the current investigation.

Reviewer question

They should present the position of the players in the pitch (goalkeeper, defender, etc);

Author responses:

Thank you for your pertinent remark.

Our football team is made up of 38 players (3 goalkeepers and 35 field players). During the intervention, we divided the team into three groups (LP, UP and CG), and each group consists of a single goalkeeper and field players (i.e. the three goalkeepers are divided into the three groups). For field players we did not mention the playing position because each player was allowed to play at least 2 or 3 playing positions (For example a left back little played a left half or an attacker. Even if when divide the group into three main positions (defender, midfielder and striker), most of the soccer player play in an offensive post and in a defensive post. For these reasons, we have eliminated this parameter and this is considered among the limitations of this work.

We added in the text:” a team of experienced elite male soccer players (38 players; 3 goalkeepers and 35 field players) was assigned randomly”

Reviewer question

They should present the time of play that each group had, because it's 90' per week, about 900' in total;

Author responses:

Thank you for your relevant remark. The following paragraph was added to “participants” section:

All three groups (LP, UP and CG) belong to the same football team. All participants were engaged in soccer training 6-7 times per week, played one official game per week from the beginning of the competitive season (September) until the end of the trial (March), and also had a weekly two hours of physical education course. Standard training sessions lasted 90-100 minutes; emphasizing skill activities at various intensities, offensive and defensive strategies, and 25 to 30 minutes of soccer play with only brief interruptions by the coach. Each Sunday, the team played an official match, during which may be play between 11 and 16 players according to the playing time of each player. The players who have not played the official match will be played a friendly match against another team or a friendly match between them, and this match will be either the same day as the official match or the next day ie Monday. This allows us to rectify and keep the same training load for all players.

Reviewer question

The tests should be taken in a grass surface instead of a tartan surface.

 Author responses:

All training sessions and weekly competition were performed on a tartan surface. For these raisons we do not perform test in grass surface.

Reviewer question

In opinion it's a very good article, well written, interesting, very applied and with a lot of interest for both researchers and coaches.

Author responses:

Thank you very much for your opinion, for your relevant remark and it gives me pleasure that you accepted to improve this work.

Reviewer 3 Report

Very quality and helpful paper for the sport of soccer. I have some comments below... Very well done intro Please add more detailed info for subject backgrounds – e.g. for their ages how good are they? Line 94 what does “elite junior” mean? Worth noting the term “elite” was not used in the subjects info earlier Line 66 explain how this was verified Line 67 what was the verbal command? Was there a verbal command? While I think the intro is well done it may worth hitting on some of these tests in the intro (e.g. the adaptations assessed by these measurements and why they are important for soccer) Line 122 how was pain etc assessed?

Author Response

Responses to reviewer-3 are highlighted in Pink

Comments and Suggestions for Authors

Reviewer question

Very quality and helpful paper for the sport of soccer. I have some comments below... Very well done intro

Author responses:

 Thank you for your pertinent remark.

Reviewer question

Please add more detailed info for subject backgrounds – e.g. for their ages how good are they?

Author responses:

-We added more information about backgrounds of subjects

Reviewer question

Line 94 what does “elite junior” mean? Worth noting the term “elite” was not used in the subjects info earlier

Line 66 explain how this was verified

Author responses:

- These players could be considered as “Elite” because many participants are member of the national team and they were participating in the first national league championship. In addition, the current players participated in ALKASS international world cup U17 and they have been ranked fourth in the world. We added « elite » in the text. 

Reviewer question

Line 67 what was the verbal command? Was there a verbal command? While I think the intro is well done it may worth hitting on some of these tests in the intro (e.g. the adaptations assessed by these measurements and why they are important for soccer)

Author responses:

Introduction is now modified as requested by reviewer-1 and reviewer-2.

 Reviewer question

Line 122 how was pain etc assessed?

Author responses:

-Pain was assessed by physical coach and the medical staff

-No injuries were made during the training period.

-It is mentioned in the result part.
